# Effects of nutritional supplements on the re-infection rate of soil-transmitted helminths in school-age children: A systematic review and meta-analysis

Aisha Ummi Jibrin Isah[1], Obinna Ikechukwu Ekwunife[2]*, Izuchukwu Loveth Ejie[2], Olena Mandrik[1]

1 School of Health and Related Research (ScHARR), The University of Sheffield, Sheffield, United Kingdom,
2 Department of Clinical Pharmacy and Pharmacy Management, Nnamdi Azikiwe University, Awka, Nigeria

* oi.ekwunife@unizik.edu.ng

## Abstract

### Background

The effect of nutritional supplements on the re-infection rate of species-specific soil-transmitted helminth infections in school-aged children remains complex and available evidence on the subject matter has not been synthesized.

### Methods

The review included randomised controlled trials (RCTs) and cluster RCTs investigating food supplements on school-aged children between the age of 4–17 years. A search for RCTs was conducted on eight databases from inception to 12th June 2019. Cochrane Risk of Bias tool was used to assess the risk of bias in all included studies. Meta-analysis and narrative synthesis were conducted to describe and analyze the results of the review. Outcomes were summarized using the mean difference or standardized mean difference where appropriate.

### Results

The search produced 1,816 records. Six studies met the inclusion criteria (five individually RCTs and one cluster RCT). Four studies reported data on all three STH species, while one study only reported data on *Ascaris lumbricoides* infections and the last study reported data on only *hookworm* infections. Overall, the risk of bias in four individual studies was low across most domains. Nutritional supplementation failed to statistically reduce the re-infection rates of the three STH species. The effect of nutritional supplements on measures of physical wellbeing in school-aged children could not be determined.

### Conclusions

The findings from this systematic review suggest that nutritional supplements for treatment of STH in children should not be encouraged unless better evidence emerges. Conclusion

**Data Availability Statement:** All relevant data are within the manuscript and its Supporting Information files.

**Funding:** The authors received no specific funding for this work.

**Competing interests:** The authors have declared that no competing interests exist.

of earlier reviews on general populations may not necessarily apply to children since children possibly have a higher re-infection rate.

## Introduction

Soil-transmitted helminths (STHs) are considered the most prevalent of the Neglected Tropical Diseases (NTDs) [1]. STHs are worms transmitted through soil contaminated with faecal matter. These worms cause infections due to lack of sanitation typically resulting from the practice of open defecation, lack of hygiene such as hand washing or walking barefoot on contaminated soil (for hookworm infection) [2,3]. The three main species of STH are roundworm (*Ascaris lumbricoides)*, the whipworm (*Trichuris trichiura)* and hookworm (*Ancylostoma duodenale* and *Necator americanus*). The infections caused by these worms occur largely in impoverished rural areas of sub-Saharan Africa, Latin America, Southeast Asia, and China [4]. For instance, the global prevalence of *Trichuris trichiura* and *hookworm* reaches 790 million and 740 million respectively, and sub-Saharan Africa and China account for over 50% of hookworm prevalence [5]. The global prevalence of *Ascaris lumbricoides* is over 1.2 billion, with China accounting for over 50% of cases [5].

STH infections are known to affect all age groups. However, school-age children, particularly of the low-income communities, are the most vulnerable to infections due to poor nutrition, inadequate sanitation, and other factors that favour the survival of the parasites [6–8]. The health consequences of STH infections may plunge children further from low-income neighbourhoods into poverty since infected children possibly have worse school performance [8].

Given that infection intensity determines the severity of morbidities associated with STH infections, the treatment approach to STH infection is periodic drug treatment (deworming) to all children living in endemic areas with albendazole (400mg) or mebendazole (500mg) [9]. Specifically, drug treatment is recommended once a year if the prevalence of STHs is over 20% and twice a year if the prevalence of STHs is over 50% [9]. Besides, health and hygiene education, as well as sanitation, is recommended as part of the STH control strategy. However, several previous studies have reported rapid rates of re-infection with STH infections soon after treatment, with roundworm and whipworm infections reoccurring in less than a year [6,10,11].

Malnutrition by reducing the effectiveness of the immune response may increase susceptibility to SHT infections [12,13]. Thus, nutritional supplementation has been regarded as a feasible means of controlling the morbidity of STH infections [14], considering that appropriate consumption of nutritional supplements plays a critical role in building up immune defences against pathogens [15]. A strong immune system may potentially decrease infection intensity and consequently the chances of re-infection. An additional attraction of nutritional supplementation is their assumed safety over an extended period, an important parameter in pediatric treatments.

Previous systematic reviews have concluded that improved nutritional status of individuals can be useful in reducing STH infections through nutritional supplementation intervention [16,17]. However, these reviews did not address several issues. Firstly, the effect of a nutritional supplement on reducing re-infection rates of STH in school-aged children was not explored. Secondly, the effect of nutritional supplements on the re-infection rate of different worm

species types was not established. Lastly, the length of the follow-up period after administering nutritional supplementation needed to observe recovery from the infection was not assessed.

Our systematic review mitigated some of these shortcomings by focusing on the effects of nutritional supplements on species-specific re-infection rates in school-aged children assessed within different follow-up periods. School-aged children were chosen specifically since the burden of STH is high in this age group.

## Aim

This systematic review aimed to assess the effects of nutritional supplements on the re-infection rates and infection intensity of different STH species in infected school-age children. Considering an assumed link between malnutrition, education performance, and worm infections [7,8], the review also explores whether there is published evidence on the impact of nutritional supplements on nutritional status and education-related outcomes.

## Methods

This systematic review, based on a pre-defined protocol, was prepared in line with the Preferred Reporting Items for Systematic Reviews and Meta-analysis (PRISMA).

### Eligibility criteria

The review included randomised controlled trials (RCTs) and cluster RCTs on school-aged children and adolescents defined by the World Health Organization as 5–17 years old [8]. Since we wanted to investigate the direct cause-effect relationship between the interventions and study outcomes, we limited the review to studies of this design to minimize the risk of confounding factors. Trials investigating nutrition and or supplements, diet and food for the treatment of STH infections, such as fortified vitamins, multi-micronutrients, minerals, sugars, iron, sodium and iodine, were eligible [18]. Included trials were those with a follow-up period of at least three months to enable assessment of long-term effects. Only studies written in the English language were included.

### Study outcome

The primary outcomes of this review were infection rate of each STH species and infection intensity at different follow-up periods [19]. The secondary outcomes were indicators of nutritional status (weight (kg), mid-upper arm circumference (MUAC-for-age)), school attendance and school productivity. We included markers of nutritional status as a secondary outcome because it is one of the main risk factors and consequences of STH infection development [1,20].

### Search strategy

The search strategy was developed together with a qualified librarian. Initially, we carried out a scoping search in the Scopus database to identify relevant keywords for the final search strategy. Scopus database was searched using a combination of "soil-transmitted helminths" and "nutrition", and the first hundred results were screened. Besides that, a query search was conducted on PubReminer search tool by combining "soil-transmitted helminths", "nutrition" and "reinfection" to identify MeSH terms, the most used words in titles and abstracts and the most active authors in the subject field [21].

The identified search terms were used to systematically search Medline (via OvidSP), CENTRAL (via Cochrane Library), EMBASE, and EBSCO on 19th April 2019 and African Index

Medicus (AIM) on 12<sup>th</sup> June 2019 (See Appendix 1) from inception. Additionally, we searched grey literature using ClinicalTrials.gov and EBSCO to identify ongoing or unpublished studies. The final search strategies for the systematic search were minimally adjusted for the compatibility of each database (Appendix 1). We imported the final search results into EndNote® online via Web of Science to organize references and identify duplicates. Afterwards, we thoroughly screened the reference lists of all included studies.

## Screening

The study selection process followed the PRISMA flow diagram presented in the results section (Fig 1). All the titles/abstracts were liberally screened by the first author (including the abstracts unless there was a high confidence that exclusion criteria were not satisfied) with the full texts screened by two authors independently. Any disagreements were solved by consensus.

## Data extraction

Following recommendations of the Cochrane Handbook for Systematic Reviews of Intervention [22], we designed a data extraction form in Microsoft Excel (version 16.27), validated it by the second reviewer, and piloted on a sub-sample of studies. The first author (AI) extracted data into the validated form, enlisting titles of included studies, study characteristics, and outcomes of interest at each of the follow-up periods. The second or last authors verified the extraction accuracy. Continuous outcome data included prevalence rates or weight (kg), the mean, standard deviations (SDs) and any other reported summary statistics such as medians and inter-quartile range. Where SDs were not available, they were derived from absolute p-values and sample sizes reported within the studies as described in the Cochrane handbook [23].

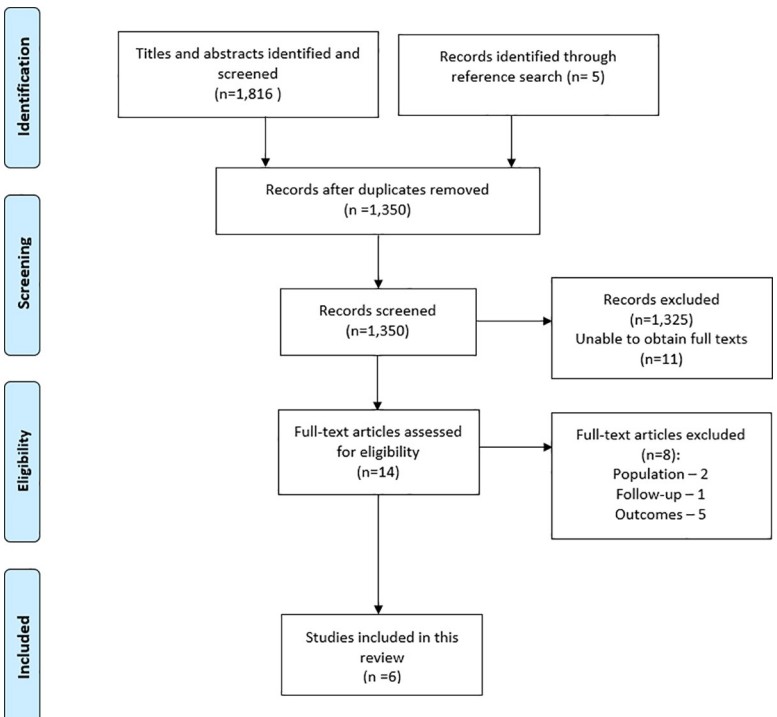

**Fig 1. PRISMA flow diagram showing search results.**

Dichotomous outcomes included the number of participants with events in both treatment and control groups and their total sample sizes.

## Assessment of risk of bias in individual studies

We used the Cochrane tool to assess the risk of bias (RoB) in original evidence [24]. Two reviewers (AI and OE) independently assessed the risk of bias of all included studies following the criteria from the Cochrane RoB tool. The domains for the assessment using this tool include random sequence generation, allocation of sequence concealment, blinding of participants and personnel, incomplete outcome data, selective outcome reporting and other potential sources of bias [25]. Each domain was assessed based on the judgment of "low risk of bias", "high risk of bias" and "unclear risk of bias", followed by a supporting statement underlying each judgment [26].

As this review included cluster RCTs, we assessed additional sources of bias under methodological heterogeneity. These sources of bias included recruitment bias, baseline imbalance, loss of clusters, incorrect analysis and comparability with RCTs [27].

## Assessment of heterogeneity

We assessed methodological heterogeneity across the included studies by exploring the differences in the underlying factors leading to statistical heterogeneity such as study designs, length of follow-up, risk of bias and reported outcomes between studies. A meta-analysis employing the random-effect model was used to assess clinical heterogeneity across studies to provide a more conservative estimate and minimise bias within studies [28]. Additionally, when clinical and methodological variations were too high across studies to combine the estimates, we used a narrative synthesis to summarize the results of the intervention effects [29].

Where meta-analysis was possible, we considered statistical heterogeneity by inspecting forest plots and $I^2$ statistic values. $I^2$ values of 0% indicated no heterogeneity, values less than 50% —moderate heterogeneity, and values greater than 50%—substantial heterogeneity in synthesized outcomes [30].

## Summary measures of treatment effects

Where meta-analysis is possible, the protocol included the following summary measures: (a) the mean difference (MD) for continuous outcomes reported on the same scale, standardized mean difference for outcomes reported on different scales, and risk or odds ratios for dichotomous data. Statistical significance was considered for outcome measures with a p-value <0.05. We used recommendations of the Cochrane Handbook to interpret the size of the effect of interventions: values of 0.2, 0.5 and 0.8 represented small, moderate and large effects respectively [31].

## Method of analysis

We used Review Manager (RevMan) 5.3 to analyze trials' data, categorized by baseline prevalence of infection (high/moderate/low) and infection intensity (heavy/moderate/light) for each of the three STH species separately [32]. For similar nutritional interventions, the results for each STH species infection were pooled separately. Each trial was presented at the longest reported follow-up periods for each STH species. Considering the limited number of studies retrieved we included in meta-analysis both RCTs and cluster-RCTs, while analyzing an impact of exclusion of cluster RCTs from the quantitative synthesis. We used a narrative synthesis to summarize the results that could not be included in the meta-analysis.

Due to the observed variations in the length of follow-up across individual studies, the effect of length of follow-up on the intervention's effect was explored in a subgroup analysis. For studies that reported multiple treatment groups, we combined the multiple treatment groups into one to create a single-pairwise comparison as recommended by the Cochrane handbook, where pooling of results was required [33].

## Subgroup analysis

Subgroup analysis investigated how the length of follow-up affects the re-infection rates of each STH species across studies. The follow-up periods to be analysed using random-effects meta-analysis included 3–5 months, 6–9 months and 10–12 months. Also, subgroup analyses for weight, MUAC-for-age, school attendance, and school productivity outcomes were planned to be conducted.

## Sensitivity analysis

Sensitivity analysis examined the effect of including trials that were cluster-randomized on the pooled effects. Another sensitivity analysis evaluated any changes in intervention effects using a fixed-effects model compared to the random-effects model.

## Result

### Search results and study selection

The literature search identified 1,816 studies from all electronic databases. Five other studies were identified from screening the reference lists. After deleting the duplicates, 1,350 studies remained. Details of the study selection process are presented in a PRISMA flow diagram (Fig 1).

### Characteristics of the included studies

A total of six studies were included in this review. Five of the studies included in this review were RCTs except for one, which was a cluster RCT [34]. There was variability in the follow-up data used within trials. While the most common follow-up period was three and six months, the longest follow-up period across all studies was up to twelve months (Table 2).

The trials were published between 2000 and 2016, each conducted in five different countries: Kenya, Zambia, Sri Lanka, Malaysia, and Cambodia. The six trials with sufficient reporting included participants with 4,272 children at baseline. All trials also included male and female participants, with a majority of them being female. The participants' ages ranged between 7–18 years. However, one of the trials did not report any age range of its participants [35].

All trials included children screened for at least one of the worm infestations, regardless of their intestinal worm load. Only three trials reported data for all three STH species (Table 1).

### Interventions

All trials assessed different nutritional supplementation interventions, and none of the trials used the same dosage. The most commonly used intervention, analysed in three studies, was iron supplements. One study used vitamin A supplement which was supplemented with micronutrient fortified food [36]. In four out of six included trials, participants in both arms received either single-dose albendazole or mebendazole before and/or after nutritional supplements were administered [34,35,37] (Table 1).

**Table 1.** Characteristics of included studies and interventions.

| Author | Country | Study Design | Study aim | Sample size | Intervention group | Control group | The age range of participant (years) | Loss to follow-up | Type of soil-transmitted helminth species | Outcomes of interest |
|---|---|---|---|---|---|---|---|---|---|---|
| De Gier et al., 2016 | Cambodia | A double-blinded, cluster-randomised, placebo-controlled trial | To study the effects of micronutrient-fortified rice on *hookworm* infection in Cambodian schoolchildren | 1,257 | Group 1: UltraRice_original Group 2: UltraRice_improved Group 3: NutriRice All participants received additional school meals including canned fish, vitamin A +D fortified vegetable oil, yellow split peas and iodized salt. All participants received 500mg of mebendazole after baseline data collection | Placebo Rice All participants received additional school meals including canned fish, vitamin A+D fortified vegetable oil, yellow split peas and iodized salt. All participants received 500mg of mebendazole after baseline data collection | 7–14 | Not reported | *Hookworm* (no distinction made between the two species) | Improve in STH infection status and cognition for children receiving fortified multi-micronutrients |
| Al-Mekhlafi et al., 2014 | Malaysia | A randomised, double-blinded, placebo-controlled trial | To assess whether vitamin A supplementation can protect children from acquiring or developing STH infections | 250 | Vitamin A(200,000 IU) supplements followed up by two pieces of fried banana (rich in oil). All participants received a 3-day course of 400 mg/daily albendazole after baseline data collection | Placebo. All participants received a 3-day course of 400 mg/daily albendazole after baseline data collection | 7–12 | 35 | *Ascaris lumbricoides, Trichuris trichiura, hookworm* | Reduction of STH infections in children receiving vitamin A |
| Ebenezer et al., 2013 | Sri Lanka | A prospective, placebo-controlled cluster randomised study | To assess the impact of deworming and iron supplementation on the cognitive abilities of school-age children in Sri Lanka | 1,190 | Iron supplementation (200mg ferrous sulphate equivalent to 60mg of elemental iron) All participants received 500-mg single-dose mebendazole after baseline testing | Placebo | Reported as school-aged children (age range not reported) | 431 | *Ascaris lumbricoides, Trichuris trichiura, hookworm* | Reduction of the prevalence of STH infections and cognition in children receiving deworming and iron supplements |
| Nchito et al., 2009 | Zambia | Randomised, placebo-controlled, double-blind, two-by-two factorial intervention trial | To determine the effect of iron and multi-micronutrients on reinfection with *Ascaris lumbricoides* | 378 | Group 1: placebo/multi-micronutrient group Group 2: Iron/placebo group Group 3: Iron/multi-micronutrient group | Placebo | 7–15 | 163 | *Ascaris lumbricoides, Trichuris trichiura, hookworm* | Reduction in STH reinfections in children receiving iron supplements and multi-micronutrients |

(*Continued*)

Table 1. (Continued)

| Author | Country | Study Design | Study aim | Sample size | Intervention group | Control group | The age range of participant (years) | Loss to follow-up | Type of soil-transmitted helminth species | Outcomes of interest |
|---|---|---|---|---|---|---|---|---|---|---|
| Olsen, 2003 | Kenya | Randomized, placebo-controlled, double-blind, two-by-two factorial trial | The effect of multimicronutrientsupplementation on reinfection with intestinal helminths and *Schistosoma mansoni*. | 997 | multimicronutrient supplementation and multihelminth chemotherapy. All infected participants received a single dose 600mg albendazole after baseline | Placebo. All infected participants received a single dose 600mg albendazole after baseline | 8–18 | Not reported | *Ascaris lumbricoides, Trichuris trichiura, hookworm* | Reduction of STH reinfections and reinfection intensity in children receiving multi-micronutrients |
| Olsen', Nawiri' and Friis, 2000 | Kenya | Randomised, placebo-controlled double-blind iron supplementation trial | To determine the effects of iron on reinfection rates and intensities of hookworm, *Ascaris lumbricoides, Trichuris trichiura, and Schistosoma mansoni* | 200 | Green film-coated ferrous dextran (200 mg corresponding to 60 mg elemental iron). All participants received a 3-day course of 400 mg/daily albendazole after baseline data collection | Placebo. All participants received a 3-day course of 400 mg/daily albendazole after baseline data collection | 7–15 | 30 | *Ascaris lumbricoides, Trichuris trichiura, hookworm* | Reduction of STH reinfections and infection intensities in children receiving iron Supplements |

## Comparators

Each trial had a placebo group as a comparator (Table 1). One study gave additional fortified food to its placebo group [34], and in two studies, additional doses of mebendazole were given to the placebo groups [34,35].

## Outcome measures

All studies reported data separately for each STH species. Three studies measured the actual re-infection rates as percentages with corresponding 95% confidence intervals (CIs), prevalence rates and infection intensity [36–38]. Two studies reported only the prevalence rates and infection intensities to define the strength of re-infection for each STH species [34,35].

Secondary outcomes were not reported in the majority of the included studies. No study reported weight changes, MUAC-for-age, or school productivity (standard test performance), and only one study reported data on school attendance [35].

## Risk of bias of individual studies

The risk of bias in included trials is presented in Fig 2. The overall risk of bias in five included studies was low, though one study was considered of unclear risk of bias because of poor reporting. Attrition bias due to missing data and other biases were the main threats for the validity of the outcomes.

## Effect of nutritional supplements

All the studies except one (Ebenezer et al., 2013) were included in the meta-analysis to summarise the re-infection rate among children. The exclusion of this study was to avoid the result being confounded by the effect of deworming since the study added deworming in the intervention arm but not in the control arm. Infection intensity was summarised using narrative synthesis for all included studies due to the presence of high statistical heterogeneity in the reported outcomes and missing relevant data (See Table 2 for further details).

A narrative synthesis was used to report the results of single trials on the use of vitamin A.

**1. Effect of nutritional supplements on re-infection rates of Ascaris lumbricoides.** *Iron*. Two studies reported sufficient data to assess a pooled effect of using iron supplements to decrease *Ascaris lumbricoides* re-infection rate (Fig 3) [37,38]. The meta-analysis of these studies did not report statistically significant effectiveness of the intervention with the average effect size of 1.00 (-13.61, 15.62).

*Vitamin A*. Only one study reported the use of vitamin A supplements to decrease the re-infection rate of *Ascaris lumbricoides* infection, hence a meta-analysis was not possible. While the authors reported a slight decrease in the re-infection rates of *Ascaris lumbricoides* at three months of follow-up in the treatment group compared to the control group, this difference was not significant at six months (p-value = 0.453) [36]. Thus, the author suggested that the observed effects were a result of the antihelminthic drugs administered after baseline data were collected. Furthermore, the study reported that rates of re-infection had fallen back to the baseline rates towards the end of the intervention [36].

*Multi-micronutrients*. Two studies reported using multimicronutrients as a single intervention to reduce re-infection rate of *Ascaris lumbricoides* infections (Fig 4) [38,39]. These studies were not able to prove the effectiveness of the intervention with the average effect size of -0.30 (-5.23, 4.63).

**2. Effect of nutritional supplements on re-infection rates of Trichuris trichiura.** *Iron*. Two studies reported using iron supplements as an intervention to decrease re-infection rates

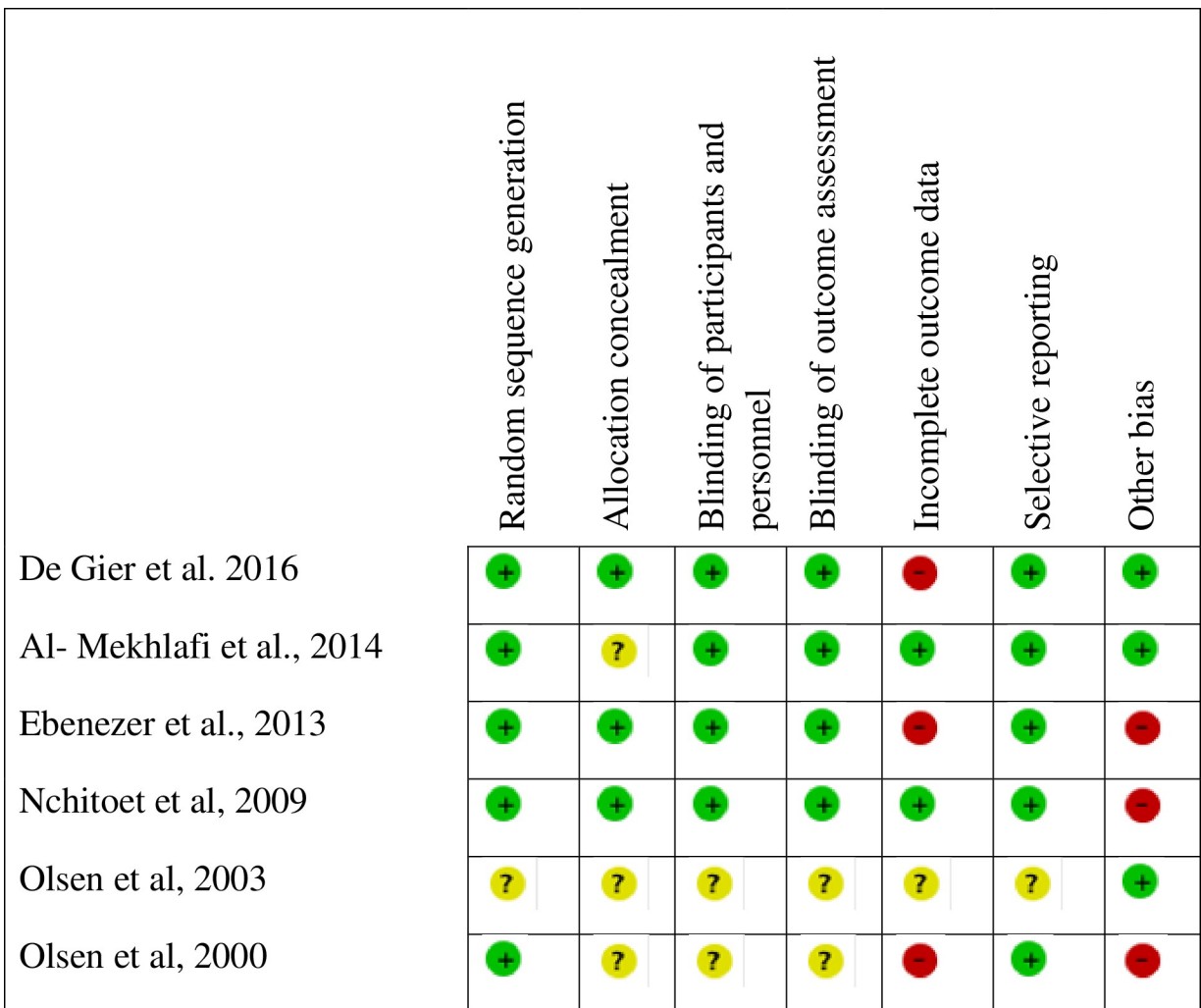

**Fig 2. Risk of bias of included studies and risk of bias summary.**

of *Trichuris trichiura* infection [35,37]. However, a statistically significant effect was reported only by cluster RCT of Ebenezer et al (2013), in which an iron supplement was assessed in combination with mebendazole.

*Vitamin A*. Only one study analysed effects of vitamin A supplement [36], concluding on no significant difference in the prevalence rate of *Trichuris trichiura* between the two treatment groups at six months follow-up (p>0.05). The study, however, reported a slight decrease in the re-infection rate with *Trichuris trichiura* at three but not the six months of the follow-up. The comparisons of the re-infection rates with baseline values and interpretation of the findings by the authors for this infection were identical to those on *Ascaris lumbricoides*. The infection intensities between the two intervention groups were also similar at follow-up and were described as heavy (p = 0.847).

*Multi-micronutrients*. Only a study with unclear risk of bias reported the use of multi-micronutrients as single interventions to decrease the re-infection rate of *Trichuris trichiura* infections [39], reporting no statistically significant difference in reinfection rates and reinfection intensities between the intervention and placebo arms.

**Table 2. Target outcomes reported by individual studies.**

| | Treatment arm | Trials (Author and year) | | | | | |
|---|---|---|---|---|---|---|---|
| | | *de Gier et al., 2016* | **Al-Mekhlafi et al., 2014** | *Ebenezer et al., 2013* | **Nchito *et al*., 2009** | **Olsen', Nawiri' and Friis, 2000** | **Olsen et al, 2003** |
| Length of follow-up (months) | | 7 | 6 | 6 | 10 | 12 | 11 |
| Prevalence of *Ascaris lumbricoides* Infection (95%CI) | Intervention arm | - | 76.8 (61.5–79.1) | 14.3 | 42.6 / 47.5 / 47.4 | 45 (35–55) | 13.1 (9.6–16.6) |
| | Control arm | - | 73.1 (60.0–77.5) | 24.5 | 46.0 | 44 (33–55) | 13.4 (9.9–16.9) |
| Prevalence of *Trichuris* trichiura Infection (95%CI) | Intervention arm | - | 65.8(53.1–68.7) | 4.9 | - | 30 (22–38) | 18.1 (13.5–22.7) |
| | Control arm | - | 66.5 (55.8–70.1) | 8.5 | - | 35 (26–44) | 20.2 (15.5–24.9) |
| Prevalence of *hookworm* infection (95% CI) | Intervention arm | 21.2 | 56.4 (30.8–69.1) | 8.0 | - | 35(25–45) | 19.9 (15.2–24.6) |
| | Control arm | 11.9 | 51.9 (30.5–65.0) | 8.7 | - | 36(26–46) | 17.6 (13.3–21.9) |
| Infection Intensity of *Ascaris lumbricoides* | Intervention arm | - | 14,867 (5,246)[2] | 85.7[3] | 2628/ 1902/ 3110[4] | 3745 (94–10 990)[5] | 2.2 (1.7–2.7) |
| | Control arm | - | 13,368 (6,367)[2] | 75.5[3] | 2646[4] | 5905 (95–15 958)[5] | 2.4 (1.9–3.0) |
| Infection Intensity of *Trichuris trichiura* | Intervention arm | - | 2,859 (799)[2] | 95.1[3] | - | 38 (18–104)[5] | 2.0 (1.6–2.3) |
| | Control arm | - | 3,678 (562)[2] | 91.5[3] | - | 30 (13–108)[5] | 1.9 (1.6–2.2) |
| Infection Intensity of *hookworm*[1] | Intervention arm | - | 14(11)[2] | 92.0[3] | - | 40 (10–110)[3] | 1.9 (1.6–2.3) |
| | Control arm | - | 11(8)[2] | 91.3[3] | - | 45 (15–80)[5] | 1.9 (1.6–2.2) |
| Weight (kg) | Intervention arm | - | - | - | - | - | - |
| | Control arm | - | - | - | - | - | - |
| Mid Upper Arm Circumference for age (MUAC-for-age) | Intervention arm | - | - | - | - | - | - |
| | Control arm | - | - | - | - | - | - |
| School attendance | Intervention arm | - | - | 61.5 | - | - | - |
| | Control arm | - | - | 61.0 | - | - | - |
| School Productivity | Intervention arm | - | - | 49.6(27.1)[2] General math test; 55.1 (25.9)[2] General Tamil test | - | - | - |
| | Control arm | - | - | 48.9(27.5)[2] General math test; 56.2 (27.1)[2] General Tamil test | - | - | - |

All outcomes are presented as reported in the original study.

NA = Not Assessed

Prevalence of all STH species Infection reported as percentages rates with 95%CI unless stated otherwise.

Percentage prevalence of infection (SD[1])

Mean (SD[2])

Percentage infection intensity (the author did not report any summary statistic[3])

Mean eggs per gram of faeces (the author did not report any summary statistic[4])

Mean eggs per gram of faeces (interquartile range[5])

**3. Effect of nutritional supplements on re-infection rates of hookworm.** *Iron*. An RCT and a cluster-RCT reported the use of iron supplements as an intervention to decrease the re-infection rate of hookworm infection [35,37]. Both studies failed to show a decrease in

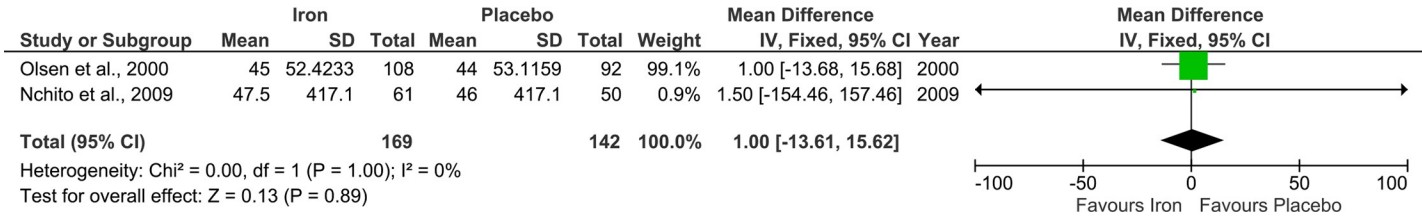

**Fig 3. Effect of iron supplements on the re-infection rate of *Ascaris lumbricoides* infection.** Outcomes represent the re-infection rate (%) at follow-up.

re-infection rates of *hookworm* infections in the treatment group compared to the control group.

*Vitamin A*. One study reported the use of vitamin A supplements on re-infection rates with *hookworm* infections. Hence not enough data was available to perform a meta-analysis. In the study, no significant differences were reported in the re-infection rates between the two treatment groups at six months follow up (p-value = 0.411). Overall, the limited quantity of available evidence suggests that vitamin A does not decrease the re-infection rate of hookworm. Furthermore, the length of follow-up in this study was not sufficient to determine if the intervention has a lasting effect on the re-infection rates.

*Multi-micronutrients*. Two trials reported the use of multi-micronutrient (fortified rice and multi-micronutrient tablets) as a treatment approach to decrease the re-infection rate of hookworm infections (Fig 5) [34,39]. As shown in Fig 5, both studies failed to show a decrease in re-infection rates of *hookworm* infections in the treatment group compared to the control group reporting an average effect size value of 0.18 (0.06,0.30).

**4. Effect of nutritional supplements on weight and MUAC-for-age.** None of the included studies assessed weight and MUAC-for-age of the study participants as an outcome measure.

**5. Effect of nutrition on school attendance and school productivity.** Only one study assessed and reported school attendance of its participants. The results reported no difference in school attendance at follow-up between the two intervention groups (Table 2). No study included in this review assessed school productivity of its participants.

## Adverse outcomes

None of the included studies reported data on any adverse outcomes.

## Supplementary analysis

For all three STH species, the effect of the nutritional supplements at the different follow-up periods, i.e. 3 to 12 months were summarised as inconclusive (Figs 6–8). The effect of the nutritional supplements failed to reach statistical significance across the three different follow-

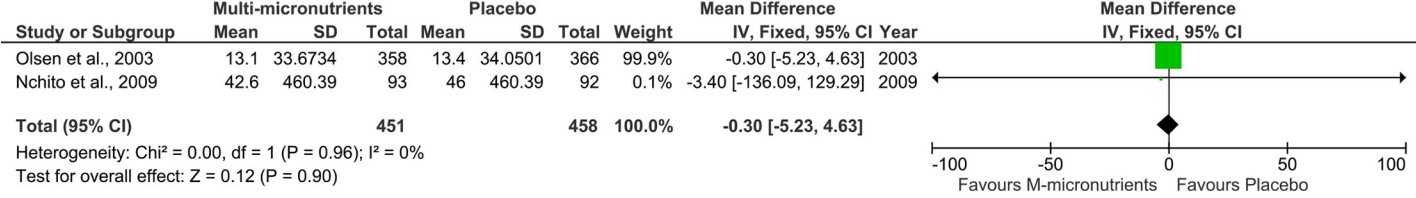

**Fig 4. Effect of multi-micronutrients on the re-infection rate of *Ascaris lumbricoides* infection.**

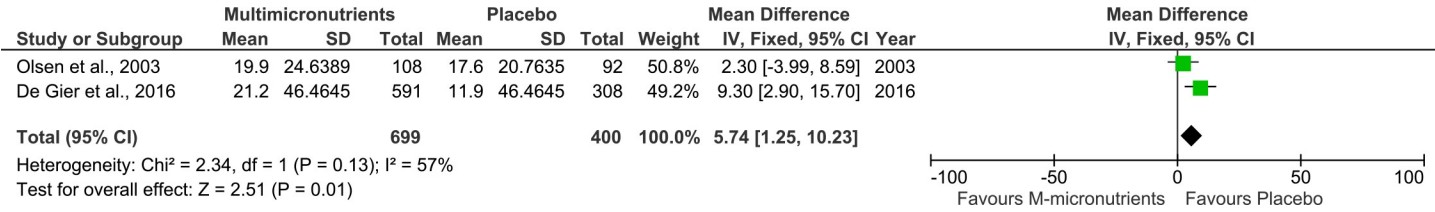

**Fig 5. Effect of multimicronutrients on the re-infection rate of hookworm infection.** Outcomes represent the re-infection rate (%) at follow-up.

up periods. It was not feasible to carry out the planned subgroup analysis by weight, MUAC-for-age, school attendance and school productivity outcomes due to insufficient data.

## Sensitivity analysis

We could not compare the results of the RCTs and the cluster-RCTs study because of an insufficient number of studies identified; expectedly, the cluster-RCT tend to report a higher impact of an intervention (iron) on STH re-infection rate. Using fixed-effect versus to random-effect model did not change the conclusions derived in this review (Figs i–k in S1 Appendix).

## Discussion

This review shows that nutritional supplements did not significantly reduce the re-infection rate of the different STH species. This effect is apparent in the observed wide confidence interval from the meta-analysis, which suggests that the effect of nutritional supplementation interventions is too small to be clinically relevant. It is also likely that the limited number of studies

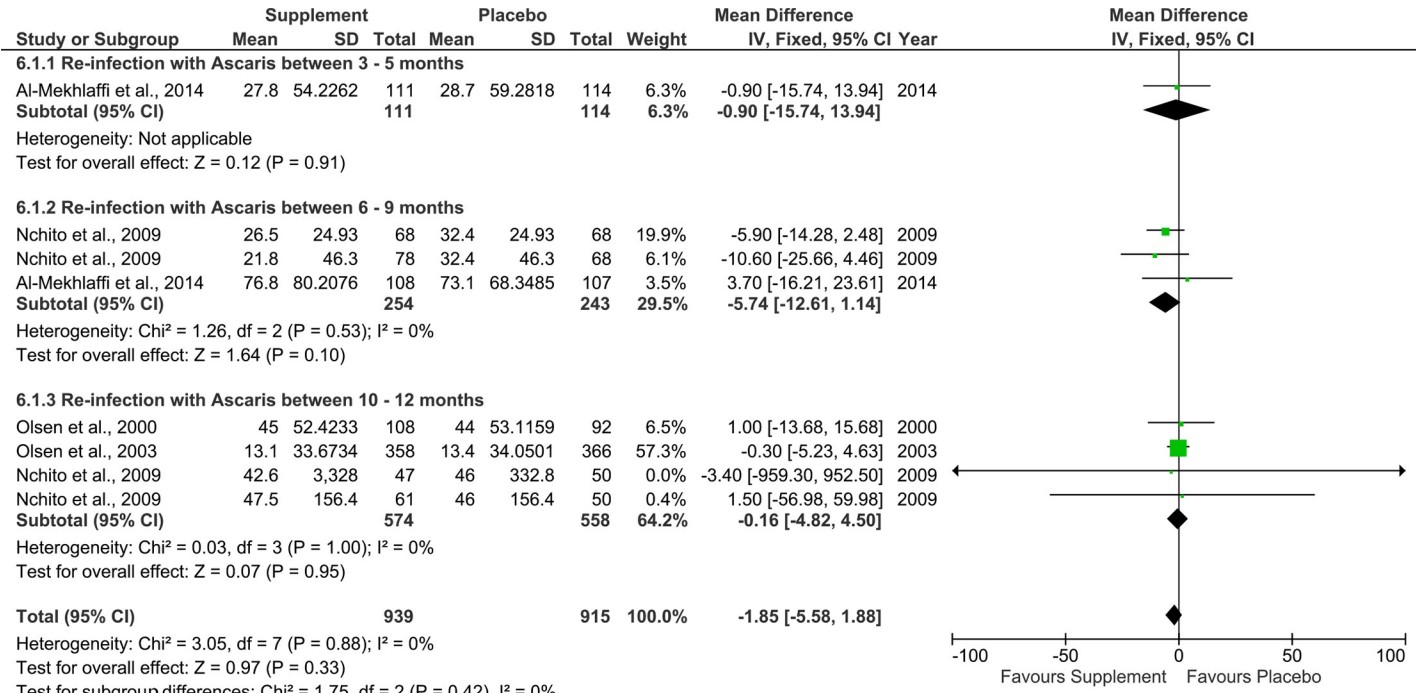

**Fig 6. Effect of nutritional supplements on re-infection rates with Ascaris lumbricoides at different follow-up periods.** Outcomes represent prevalence rates of reinfection (%).

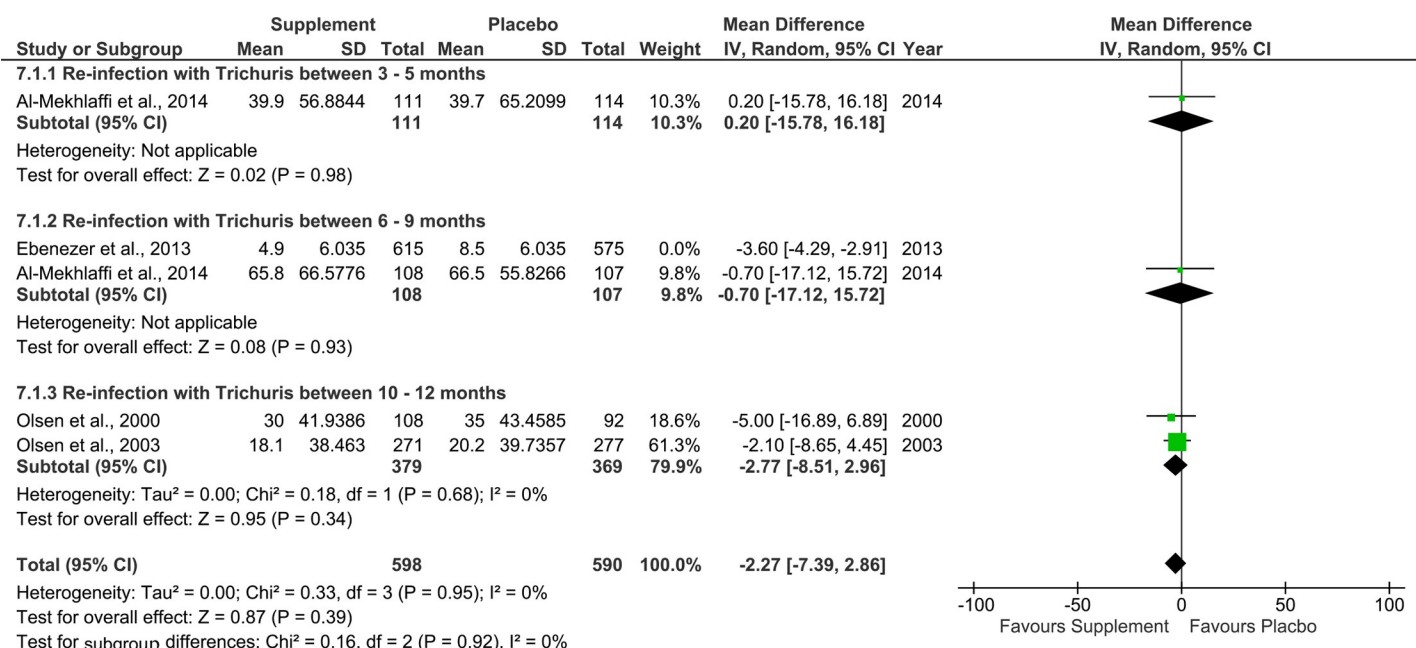

**Fig 7. Effect of nutritional supplements on re-infection rates with *Trichuris trichiura* at different follow-up periods.** Outcomes represent prevalence rates of re-infection (%).

used in this review contributed to the inconclusive effect of nutritional supplementation interventions.

While findings of our review do not encourage nutritional supplements as a deworming intervention among children, they are especially important considering the previous contradictory evidence. With children being the most vulnerable and infected group with STH

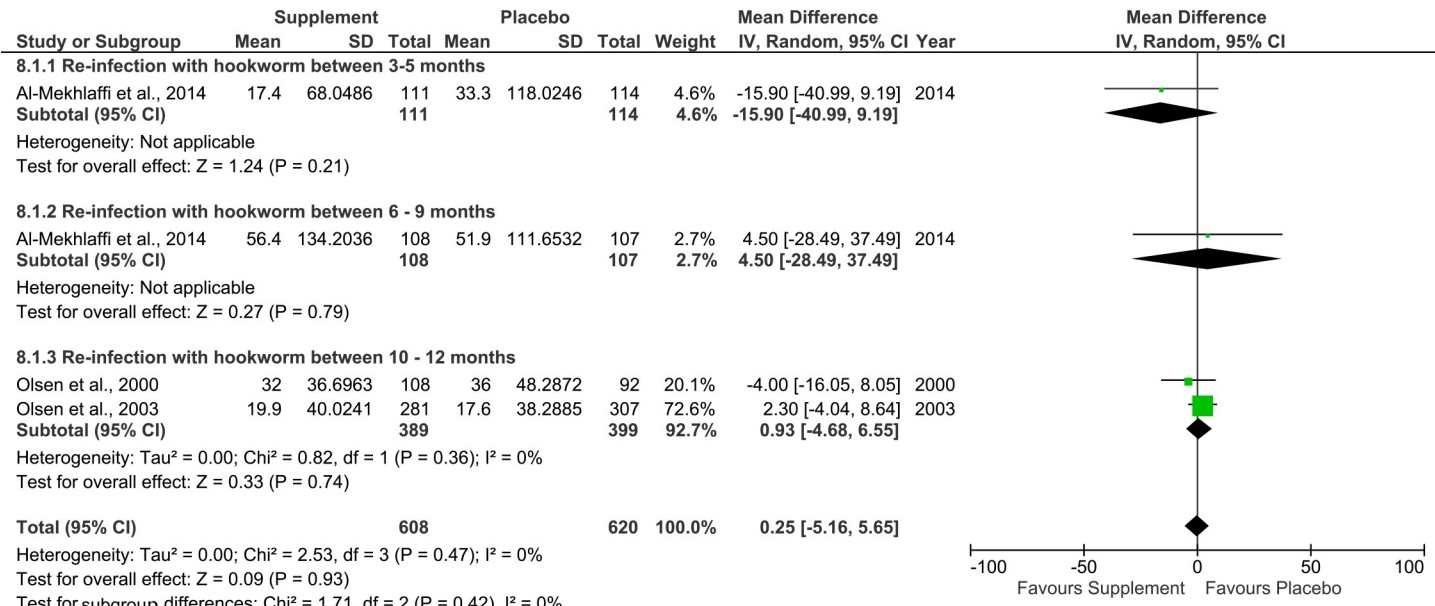

**Fig 8. Effect of nutritional supplements on reinfection rates with *hookworm* at different follow-up periods.** Outcomes represent prevalence rates of reinfection (%).

infections [40], little evidence exists on the effect of nutritional supplements on the re-infection rate of the different STH species. The World Health Organization, alongside the Food and Agriculture Organization of the United Nations and the United Nations Environment Programme, has long advocated for the need to create safe and urgent interventions to control the incidence of worm infections, particularly among children [32]. Previous studies have reported that the impact of infections is dependent on the worm load and nutrition status of the hosts [41] and that some multi-micronutrients have important effects on the developing immune system [42]. Hence, receiving the right nutrition in early lives may support building children's immune systems against any form of diseases. Our review though demonstrates that promotions of the nutritional supplements as a safe and effective intervention among children for the treatment of soil-transmitted helminthic infections require stronger supporting evidence.

The results of our synthesis are partially contradictory to a previous systematic review on this topic. Yap et al (2014) reported a positive odds ratio effect of 0.75 (0.54,1.05) of iron supplementation interventions in decreasing re-infection rates with *Ascaris lumbricoides* infection [16]. This difference in results and conclusion could be related to the differences in methodologies of both reviews. Yap et al (2014) applied broader eligibility criteria, enabled them to retrieve and include up to 15 studies (three times more than the number of studies included in this review). The population of our review was limited to only school-aged children, while the previous review considered participants of all age groups, including both pre-school and school-aged children, adolescents, and adults. Thus, the observed effects in the previous review may be as a result of the combined intervention effects in other population groups. Children groups are more likely to be prone to re-infection with STH species compared to adults, possibly because of their poor sanitation and hygiene management [43]. Besides, all the included trials in this review were conducted in rural areas, where proper sanitation practices may not exist and thus contributed to the weak effect of the nutritional supplements. Hence, reviews involving adult participants may show better effects of nutritional supplements compared to reviews using only children participants. Secondly, Yap et al (2014) considered both RCTs and prospective cohort studies in their synthesis, while this review only used RCTs and cluster-RCTs in the data analysis. Prospective cohort trials are prone to various types of bias, including selection bias, information bias, and confounding [44]. Hence, including only RCTs in this review increases the certainty in its findings. Thirdly, both reviews performed data analysis at the longest reported follow-up periods. However, in this review, additional data analysis was done at baseline and different follow-up periods. The study authors reported slightly lower re-infection rates at the first follow-up periods, but this effect diminished towards the end of the intervention follow-up periods. Besides, the previous study did not consider the infection status of its participants in its data analysis but rather their nutrition statuses. Thus, this consideration in the analysis of the previous review may have influenced the observed effect of iron supplementation on the different STH infections. Finally, the differences in the tools used in assessing the quality of the included studies may have affected the reporting of biases of the original evidence.

## Limitations

There are several limitations in both the included individual studies and the review that may have influenced the findings of this review. Despite the developed comprehensive search, we acknowledge that certain articles may be missed by this systematic review. Evidence gathered in this review did not fully explore the effect of nutritional supplements on the re-infection rate of each STH species due to the presence of methodological heterogeneity across studies. As a result of methodological heterogeneity (varying reporting scales, follow-up periods,

infection types) across studies, only four studies were included in the meta-analysis. It must be noted that applying the random-effects model to the few studies in this review can result in poor performance, leading to the observed wider CIs with compromised coverage probability [45]. Furthermore, the use of a few studies could result in poor estimation of heterogeneity between studies. Secondly, none of the studies administered nutritional supplements according to the study participant's worm loads. Participants with heavier worm loads are likely to require a larger dosage of intervention compared to those with lighter worm loads [41]. Also, participants with heavier worm loads may absorb nutrients less efficiently than those with lighter ones, since the interactions of heavy infections affect the intestinal walls [41]. With regards to the effects of a nutritional supplement on STH species, each participant was given the same type of nutrients regardless of the species infecting each individual. Thus, this may have affected the effectiveness of the interventions. Limiting this meta-analysis to school-based interventions to avoid reporting, selection, and loss to follow up bias, could also result in including a healthier population in the synthesis.

More studies are needed to provide sufficient evidence for the recommendation of nutritional supplements as a deworming strategy in school-aged children since the small number of studies included in this review did not show that supplements decrease re-infection rate in this population group. Also, each study included in this review used different types of nutritional supplementation with varying dosages. Hence, the methodological heterogeneity of the included study could have possibly contributed to the observed negative result.

The majority of the studies included in this review were RCTs reporting small sample sizes. Trials using large sample sizes have been reported to present more significant weight on intervention effects, by producing narrower CIs and more precise effect estimates compared to trials using small sample sizes [46]. Therefore, larger RCTs are needed to confirm the effect of nutritional supplements on reducing re-infection rate of STHs. This meta-analysis targeted to synthesize RCTs as the most robust evidence; considering the limited evidence retrieved, a supplementary review of quasi-experimental studies would complement the reported analysis, though in this analysis we observed the difference in outcomes reported by randomization approach (truly-randomized versus cluster-randomized).

## Implications

The finding of this review has implication for practice. Even though there is little evidence on the long-term effects of iron supplements to decrease re-infection rates of each STH species in infected children, the overall effects of nutritional supplements remain inconclusive. Also, the strength of evidence generated from this systematic review is too low to provide a base for policymakers to make recommendations at both national and international levels. Thus, a review of high-quality prospective cohort studies with a long follow-up would contribute to strengthening the conclusion on the impact of nutritional supplements on the re-infection rate of soil-transmitted helminths in children.

More studies with larger samples are needed to confirm the potential long-term benefits of nutritional supplement interventions in children infected with STH infections. Further research should also consider the specific nutrition dosage required for each type of STH infection, as the different species could require different dosages for more effective results. Even though the included trials generally had low risks of bias, they were still prone to some design flaws such as inadequate randomisation, selective reporting and unblinded outcome assessments that may lead to biases and decrease the validity of results. More rigorous methods of randomisation methods can be applied to future studies to increase the reliability and statistical significance of the intervention effects on the outcomes of interest. For example, infected

individuals have been shown to have significant differences in the intensity of infections. Hence, participants can be stratified according to their egg loads before random allocations are carried out. This may provide researchers with the bases for treatment dosage guidelines.

The included trials were mainly RCTs randomized at the individual level and with small sample sizes. However, there is lack of consensus regarding the most effective type of trial design, considering the difficulty in reaching large sample sizes in RCTs. Hence, future studies could benefit by comparing the outcomes reported in truly RCT, cluster RCTs and factorial designs, as they are feasible to use on larger sample sizes [47]. While cluster RCTs have been reported to have more potential to detect treatment effects in the most affected groups within the clusters [48,49], the risk of bias in such studies should be detailed, exploring the differences in outcomes reported. Furthermore, each type of STH species is likely to react to nutritional supplements in distinctive ways. Hence future research could also consider the implementation of nutritional supplement treatments according to the type of infection.

Finally, no RCT reported adverse events related to the use of nutritional supplements in the deworming of children. Studies powered to assess negative impacts of nutritional supplements (for instance multimicronutrients) would be valued to highlight their possible negative impact on infection rate and health in general.

## Conclusions

This systematic review is the first to investigate the effects of nutritional supplements on the strength of STH species-specific re-infection rates in school-aged children. The current evidence gathered in this review is weak to conclude that nutritional interventions had an impact on the prevalence rates and infection intensities of each STH species. Thus, nutritional supplements for treatment of STH in children should not be encouraged unless better evidence emerges. Conclusion of earlier reviews on general populations may not necessarily apply to children since children possibly have a higher re-infection rate due to hygiene.

## Supporting information

**S1 Checklist. PRISMA checklist.**
(DOC)

**S1 Appendix.**
(DOCX)

## Author Contributions

**Conceptualization:** Aisha Ummi Jibrin Isah, Olena Mandrik.

**Formal analysis:** Aisha Ummi Jibrin Isah, Obinna Ikechukwu Ekwunife, Izuchukwu Loveth Ejie.

**Methodology:** Aisha Ummi Jibrin Isah, Olena Mandrik.

**Supervision:** Olena Mandrik.

**Validation:** Aisha Ummi Jibrin Isah, Obinna Ikechukwu Ekwunife, Olena Mandrik.

**Writing – original draft:** Aisha Ummi Jibrin Isah, Olena Mandrik.

**Writing – review & editing:** Aisha Ummi Jibrin Isah, Obinna Ikechukwu Ekwunife, Izuchukwu Loveth Ejie, Olena Mandrik.

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
