## [Decision Letter · Decision Letter 0]

28 May 2020

PONE-D-20-08643

Effects of nutritional supplements on the re-infection rate of soil-transmitted helminths in school-age children: A systematic review and meta-analysis

PLOS ONE

Dear Dr. Ekwunife,

Thank you for submitting your manuscript to PLOS ONE. After careful consideration, we feel that it has merit but does not fully meet PLOS ONE’s publication criteria as it currently stands. Therefore, we invite you to submit a revised version of the manuscript that addresses the points raised during the review process.

Please pay due attention to the remarks of reviewer 1 on the discrepancy between the # of children reported in the text and Figure 3 for the Ebenezer study. 

We look forward to receiving your revised manuscript.

Kind regards,

Frank Wieringa, M.D., Ph.D.

Academic Editor

PLOS ONE

Journal Requirements:

2. Please ensure that you refer to Figures i, j and k in your text as, if accepted, production will need this reference to link the reader to each figure.

Reviewers' comments:

Reviewer's Responses to Questions

**Comments to the Author**

1. Is the manuscript technically sound, and do the data support the conclusions?

Reviewer #1: Partly

Reviewer #2: Yes

2. Has the statistical analysis been performed appropriately and rigorously? 

Reviewer #1: No

Reviewer #2: Yes

3. Have the authors made all data underlying the findings in their manuscript fully available?

Reviewer #1: Yes

Reviewer #2: Yes

4. Is the manuscript presented in an intelligible fashion and written in standard English?

Reviewer #1: Yes

Reviewer #2: Yes

5. Review Comments to the Author

Reviewer #1: -systematic review is well written, follows accepted guidelines for reporting (PRISMA) and is based on a pre-defined protocol.

-what is the “liberal accelerated approach” to screening- this needs a reference and some detail

-is the protocol referenced?

-risk of bias is well-done according to accepted Cochrane standards

-it is unclear why nutritional supplements would be expected to decrease reinfection rate of children- I see the authors have cited prior systematic reviews in this area, but I do not understand the biological/physiological rationale for why nutritional supplements would have this effect. I suggest the authors provide 2-3 sentences about why nutritional supplements are expected to have this effect

-with respect to the statistical analysis and whether data support conclusions, the authors need to consider the independent effect of deworming in these studies. It appears that 3 of the studies had deworming in the intervention arm, and two did not. Also, two studies reported additional doses of mebendazole for the placebo group (page 14, line 302). I think that studies with deworming in the intervention arms should be analyzed separately from the studies which have no deworming. Without doing this, the results may be confounded by the effect of deworming on reinfection, and the conclusions may be misleading also

For example, in the figure 3 of iron supplementation, Ebenezer 2013 is a study of iron+deworming vs no deworming, thus it is not surprising that it has a beneficial effect on reinfection rates.

Figure 3- it is unclear why the Ebenezer study has less than 300 children in this analysis when the table of included studies indicates this study has over 1000 participants. Similarly, figure 4 shows only 79 participants from the Ebenzer study. These seem to indicate that the Ebenezer study sampled only a few children to conduct STH analysis-this should be described in the description of studies somehow.

-Analysis and results- there is no assessment of publication bias.

-the main findings do not consider the strength of evidence for the interpretation of results, which is a PRISMA reporting standard. ie standard 24: Summarize the main findings including the strength of evidence for each main outcome; consider their relevance to key groups (e.g., healthcare providers, users, and policy makers).

-Results section, some statements are written in a way that presumes effectiveness, such as “The three studies were not 347 able to prove the effectiveness of the intervention with the average effect size of -0.15(-0.48, 348 0.17)”. I suggest authors review the manuscript to write results in a more neutral way.

-Discussion section “Hence, including only RCTs in this review may have contributed to its high quality.” I disagree with this sentence. I think including only RCTs may increase the certainty of findings, but does not necessarily increase the quality of the systematic review. I suggest the authors reword.

-Discussion- the discussion needs to consider that there may be other reasons to recommend nutritional supplements-ie to promote nutritional status. Thus I think the wording of sentences such as this one: “More studies are needed to provide sufficient evidence for the recommendation of nutritional supplements in school-aged children since the small number of studies included in this review did not show that nutritional supplements reduce the re-infection rate of the different STH species in school-aged children.” This suggests there are no benefits of nutritional supplementation, which seems to go beyond the review since this review focused only on reinfection rates

Reviewer #2: This is a an important manuscript, reporting important findings on the effects of nutritional supplements on re-infection rate of soil transmitted helminths. I do have some comments, please see attachment for my comments.

6. PLOS authors have the option to publish the peer review history of their article (what does this mean?). If published, this will include your full peer review and any attached files.

Reviewer #1: No

Reviewer #2: No

---

## [Author Response · Author response to Decision Letter 0]

12 Jun 2020

Reviewer #1: -systematic review is well written, follows accepted guidelines for reporting (PRISMA) and is based on a pre-defined protocol.

-what is the “liberal accelerated approach” to screening- this needs a reference and some detail

Reply: We corrected the text.

-is the protocol referenced?

Reply: The protocol was registered by the student ID 180250633 at the blackboard system of the University of Sheffield. Though since the access to the system is internal, we did not include this information as not useful for the reader.

-risk of bias is well-done according to accepted Cochrane standards

Reply: Thank you

-it is unclear why nutritional supplements would be expected to decrease reinfection rate of children- I see the authors have cited prior systematic reviews in this area, but I do not understand the biological/physiological rationale for why nutritional supplements would have this effect. I suggest the authors provide 2-3 sentences about why nutritional supplements are expected to have this effect

Reply: The rationale for this review is provided on lines 96 – 103, just before citing the previous reviews and their limitations. We included an additional sentence on relation of malnutrition and SHT infections (line 96).

-with respect to the statistical analysis and whether data support conclusions, the authors need to consider the independent effect of deworming in these studies. It appears that 3 of the studies had deworming in the intervention arm, and two did not. Also, two studies reported additional doses of mebendazole for the placebo group (page 14, line 302). I think that studies with deworming in the intervention arms should be analyzed separately from the studies which have no deworming. Without doing this, the results may be confounded by the effect of deworming on reinfection, and the conclusions may be misleading also

For example, in the figure 3 of iron supplementation, Ebenezer 2013 is a study of iron+deworming vs no deworming, thus it is not surprising that it has a beneficial effect on reinfection rates.

Figure 3- it is unclear why the Ebenezer study has less than 300 children in this analysis when the table of included studies indicates this study has over 1000 participants. Similarly, figure 4 shows only 79 participants from the Ebenzer study. These seem to indicate that the Ebenezer study sampled only a few children to conduct STH analysis-this should be described in the description of studies somehow.

Reply: We agree with the reviewer that Ebenzer study confounds the effect of deworming. We excluded this trial from meta-analysis and added the relevant statements into the Results section (lines 325-328). 

The second trial by Nchito et al 2009 does not bias the results because both intervention and control group did not have supplementary treatment while the intervention effect is measured in comparison to the control. The results of this trial are homogenous to other trials. 

Additionally, we corrected the number of participants in Ebenezer study although it was excluded in the meta-analysis. The initial mistake was because the authors presented the observed outcome as N which we mistakenly considered as number of participants. 

-Analysis and results- there is no assessment of publication bias.

Reply: Because of the limited number of studies retrieved we were not able to assess publication bias, as it is instructed by the Cochrane Handbook 5.1.:

“As a rule of thumb, tests for funnel plot asymmetry should be used only when there are at least 10 studies included in the meta-analysis, because when there are fewer studies the power of the tests is too low to distinguish chance from real asymmetry.”

-the main findings do not consider the strength of evidence for the interpretation of results, which is a PRISMA reporting standard. ie standard 24: Summarize the main findings including the strength of evidence for each main outcome; consider their relevance to key groups (e.g., healthcare providers, users, and policy makers).

Reply: 

The hierarchy of evidence places RCTs on the top of the pyramid with observational studies having much less strengths in comparison to quazi-experimental and even more experimental designs. This review included only RCTs with low risk of bias. Thus, comparison of the outcomes “by the strength of evidence” will not be applicable for this review, which is already based on strong evidence itself. Since almost no secondary outcomes were identified, we can neither detail nor conclude on impact of nutritional supplements on the secondary outcomes.

We did though considered the uncertainty in results what is reflected in the Discussion section. For instance, the lines 446-449, mentioning the wide confidence intervals of the main outcomes. We also discussed the limitations of the included studies and the review itself (lines 515-532). 

The relevance of the outcomes to key groups are reported on lines 534-538.

-Results section, some statements are written in a way that presumes effectiveness, such as “The three studies were not 347 able to prove the effectiveness of the intervention with the average effect size of -0.15(-0.48, 348 0.17)”. I suggest authors review the manuscript to write results in a more neutral way.

Reply: 

This sentence refers to the objectives and the design of the included trials. Since we included trials of superiority (and not non-inferiority) design, the presumed effectiveness refers to the null hypothesis of the trials, attempting to measure a positive effect of interventions. 

-Discussion section “Hence, including only RCTs in this review may have contributed to its high quality.” I disagree with this sentence. I think including only RCTs may increase the certainty of findings, but does not necessarily increase the quality of the systematic review. I suggest the authors reword.

Reply: We agreed with the reviewer and re-phrased this sentence. 

-Discussion- the discussion needs to consider that there may be other reasons to recommend nutritional supplements-ie to promote nutritional status. Thus I think the wording of sentences such as this one: “More studies are needed to provide sufficient evidence for the recommendation of nutritional supplements in school-aged children since the small number of studies included in this review did not show that nutritional supplements reduce the re-infection rate of the different STH species in school-aged children.” This suggests there are no benefits of nutritional supplementation, which seems to go beyond the review since this review focused only on reinfection rates

Reply: We agree with the reviewer, the sentence referred to recommendations on deworming strategies. We re-phrased this sentence. 

 

Page 4: line 72-73; For the last part of the sentence: ‘and walking barefoot on contaminated soil’ is only the case for hookworm not forthe other helminths

Reply: We added “for hookworm infection” to the text.

Page 4: line 84: ‘The health consequences of STH infections may plunge further the children from low-income neighborhoods into poverty since infected children possibly have worse school performance’

I think it would be better to write: The health consequences of STH infections maytheplunge childrenfurther from low-income neighborhoods into poverty since infected children possibly have worse school performance

Reply: We corrected the text.

Page 6: line 136: the secondary outcome: interesting that the authors also want to investigate this. However how does this relateto theresearch question? I would suggest to either add this as a sub question in the introduction or explain better how this relatedto the RQ.

Also this secondary outcome measure is not mentioned in the rest of the methods section.

Reply: We added a secondary aim to the Introduction (lines 117-119), explaining the reason behind including the secondary outcomes. The methods section reports the review’s methodology independently on the outcomes (eligibility criteria, search strategy, data extraction, etc.). The lines 265-267 state that we planned to conduct the sub-group analysis for the secondary outcomes as well, though it was not possible because of insufficient number of studies retrieved. 

Page 7: it would be good to add the exact search terms that were used and also the dates that the searches were performed.

Reply: We report the search dates on lines 152-154 and the detailed search strategy in the Appendix 1, referenced on lines 152-154. 

Page 9, figure 1: 

- ‘additional records” ; i find this is a bit vague. elsewhere in the manuscript (page 13) the authors state that these record were identified from the reference list from the selected papers. It would be better tobe consistent.

- Full text excluded with reasons; this is also too vague. Please reformulate.

Reply: The Figure 1 was corrected accordingly

Page 12: line 276; the selection process is described to superficially. It would be good to elaborate on this process.

Reply: We apologize for the mistake in this sentence. This sentence referred to a trial, included in abstract screening deviating from the protocol, though excluded afterwards because of a short follow up (<3 months). The minimum age in all included studies was >7 years (Table 1). We deleted the mentioned statement. 

Page 23: line 347: ‘the three studies were not able to prove’ . Think a better formulation would be were not able to report. Prove is too strong in my opinion.

Reply: The sentence was re-formulated.

Page 27; line 415 multi-micronutrients; this is the only meta-analysis which is statistically significant, however this is not mentioned. I wonder why? Also this meta-analysis (even though based on 2 studies) shows that MM’s might have adverse effects. I think it would be good to give important this finding more attention. Please also elaborate on this important finding in the discussion. 

Reply: We revised the statistical analysis using mean difference instead of the standardized mean difference. An average effect size was 0.18 (0.06,0.30) for hookworm infections in the treatment group compared to the control group. While there is a small favorable difference towards placebo, we consider that this finding is dangerous to interpret directly as an evidence on the harmful effect of the intervention because of the superiority design of the trials (and not non-inferiority), especially considering that only two studies were identified. 

To address this comment of the reviewer, we added a sentence to the discussion on a value of additional evidence on this regard(lines 598-601).

Page 28: line 428: effect on school productivity. 

This outcome is not mentioned before. Please also elaborate on this in the methods.

Reply: School productivity was included in secondary outcomes (line 300) and was reported in Table 2. We clarified that it means standard test performance (line 301).

For the discussion I would recommend to;

- Include the value of observationational (longitudinal) studies, as eventhough they may be more prone to confounding, they do often have longer followup time and may thusprovide more evidence. 

- Give more attention to the possibility of unfavourable effect of MM supplementation.

Reply: The requested statements were added to the Discussion section (lines 527-529). We added a sentence to the discussion on a value of additional evidence on this regard, considering that superiority design of the trials may not allow to retrieve direct conclusions on unfavorable effect of MM supplementation (lines 598-601).

---

## [Editor Report · Decision Letter 1]

21 Jul 2020

Effects of nutritional supplements on the re-infection rate of soil-transmitted helminths in school-age children: A systematic review and meta-analysis

PONE-D-20-08643R1

Dear Dr. Ekwunife,

We’re pleased to inform you that your manuscript has been judged scientifically suitable for publication and will be formally accepted for publication once it meets all outstanding technical requirements.

Kind regards,

Frank Wieringa, M.D., Ph.D.

Academic Editor

PLOS ONE
---

## [Editor Report · Acceptance letter]

24 Jul 2020

PONE-D-20-08643R1 

Effects of nutritional supplements on the re-infection rate of soil-transmitted helminths in school-age children: A systematic review and meta-analysis 

Dear Dr. Ekwunife:

I'm pleased to inform you that your manuscript has been deemed suitable for publication in PLOS ONE. Congratulations! Your manuscript is now with our production department. 

Kind regards, 

on behalf of

Dr. Frank Wieringa 

Academic Editor

PLOS ONE